# Thromboembolic and Bleeding Complications in Transcatheter Aortic Valve Implantation: Insights on Mechanisms, Prophylaxis and Therapy

**DOI:** 10.3390/jcm8020280

**Published:** 2019-02-25

**Authors:** Mark P. Ranasinghe, Karlheinz Peter, James D. McFadyen

**Affiliations:** 1Atherothrombosis and Vascular Biology Program, Baker Heart and Diabetes Institute, 75 Commercial Road, PO Box 6492, Melbourne, Victoria 3004, Australia; pimalr@gmail.com (M.P.R.); james.mcfadyen@monash.edu (J.D.M.); 2Department of Medicine, Monash University, Melbourne, Victoria 3800, Australia; 3Heart Centre, The Alfred Hospital, 55 Commercial Road, Melbourne, Victoria 3004, Australia; 4Department of Clinical Haematology, The Alfred Hospital, 55 Commercial Road, Melbourne, Victoria 3004, Australia

**Keywords:** TAVI, aortic valve stenosis, antithrombotic drugs, transcatheter aortic valve implantation, complications, bleeding, anticoagulation, antiplatelet, mechanisms, therapy

## Abstract

Transcatheter aortic valve implantation (TAVI) has emerged as an important alternative to surgical aortic valve repair (SAVR) for patients with severe aortic stenosis. This rapidly advancing field has produced new-generation devices being delivered with small delivery sheaths, embolic protection devices and improved retrieval features. Despite efforts to reduce the rate of thrombotic complications associated with TAVI, valve thrombosis and cerebral ischaemic events post-TAVI continue to be a significant issue. However, the antithrombotic treatments utilised to prevent these dreaded complications are based on weak evidence and are associated with high rates of bleeding, which in itself is associated with adverse clinical outcomes. Recently, experimental data has shed light on the unique mechanisms, particularly the complex haemodynamic changes at sites of TAVI, that underpin the development of post-TAVI thrombosis. These new insights regarding the drivers of TAVI-associated thrombosis, coupled with the ongoing development of novel antithrombotics which do not cause bleeding, hold the potential to deliver newer, safer therapeutic paradigms to prevent post-TAVI thrombotic and bleeding complications. This review highlights the major challenge of post-TAVI thrombosis and bleeding, and the significant issues surrounding current antithrombotic approaches. Moreover, a detailed discussion regarding the mechanisms of post-TAVI thrombosis is provided, in addition to an appraisal of current antithrombotic guidelines, past and ongoing clinical trials, and how novel therapeutics offer the hope of optimizing antithrombotic strategies and ultimately improving patient outcomes.

## 1. Introduction

Severe aortic stenosis (AS) represents a major global healthcare burden, with medical management being associated with a poor prognosis and high rates of mortality [1]. Calcific aortic stenosis is the most common type of valvular heart disease requiring intervention, and its prevalence is growing with the increasing global life expectancy [2]. While prosthetic surgical replacement has traditionally been the mainstay treatment, the number of patients for which this approach is appropriate has been limited given that aortic stenosis is predominately a condition of older patients who often have other medical co-morbidities [3,4,5]. The growing demand for non-surgical treatment alternatives for patients with severe AS, especially in intermediate to high-risk patients, has been one of the driving forces behind the emergence of transcatheter aortic valve implantation (TAVI) [3].

Since the first procedure in 2002, TAVI has not only offered a new approach to treating calcific aortic stenosis but has also provided the large cohort of patients deemed intermediate to high surgical risk access to life-saving treatment [6,7]. This is particularly important since up to one-third of all patients in need of an aortic valve replacement are unsuitable for surgical aortic valve replacement (SAVR) [8].

The TAVI procedure involves guiding a prosthetic valve, typically via a transfemoral catheter, and implanting it into the existing aortic valve [9]. With rapid developments in device design, coupled with the growing experience of clinicians and the adoption of more efficient techniques, TAVI has become an increasingly effective procedure with fewer complications. 

As a consequence, there have now been more than 500,000 TAVI procedures performed in over 70 countries and at over 1000 centres [10].

## 2. Basis for Antithrombotic Therapy after Implantation

In spite of these developments, the issue of ischaemic and bleeding events continues to impact the immediate and long-term morbidity and mortality of patients undergoing TAVI procedures (Table 1).

Based on the results from the Surgical Replacement and Transcatheter Aortic Valve Implantation (SURTAVI) trial, the Placement of Aortic Transcatheter Valves (PARTNER) 2 cohort and the US CoreValve studies [11,12,13,15], the percentage of stroke events within 30 days of TAVI was 4.1% (± 0.7), with life-threatening and major bleeding complications observed in 10.2% (± 3.5) of patients. Within one year, these percentages increased to 7.0% (± 1.7) and 15.95% (± 0.9), respectively. Importantly, the rate of new-onset atrial fibrillation (AF) was 11.2% (± 1.9) within 30 days, which confers an increased risk of cerebrovascular events. The all-cause mortality following TAVI was 2.6% (± 0.6) within 30 days, 7.1% (± 2.9) within one year and increased to 10.7% (± 4.1) within two years of the procedure. 

A recent meta-analysis highlighted that post-TAVI bleeding was associated with a 323% increase in 30-day postoperative mortality (odds risk (OR): 4.23, 95% confidence interval (CI): 2.80–6.40; *p* < 0.0001) and that patients with major bleeding/life-threatening bleeding showed a 410% increase in mortality compared with patients without bleeding (OR: 5.10; 95% CI: 3.17–8.19; *p* < 0.0001). In addition, the presence of atrial fibrillation was independently correlated with TAVI-associated bleeding (OR: 2.63; 95% CI: 1.33–5.21; *p* = 0.005) [16].

In addition to the bleeding risk associated with antithrombotic therapy, it is important to note that scoring systems, such as the EuroSCORE II and the Society of Thoracic Surgeons (STS) risk score, can be utilised to stratify the mortality risk of patients undergoing TAVI. These scores incorporate a range of clinical variables such as age, renal impairment and New York Heart Association Functional Classification as part of a clinically validated risk assessment model to help predict outcomes in patients undergoing cardiac procedures [17,18]. 

However, important factors that heavily influence mortality after percutaneous procedures, such as frailty and body mass index, are not included in these scoring systems [19]. Highlighting the important role these clinical variables play in predicting post-procedural mortality in the context of percutaneous coronary intervention (PCI) and TAVI, frailty is an independent risk factor associated with one-year mortality post-TAVI (hazard ratio (HR): 3.5, 95% CI: 1.4 to 8.5, *p* = 0.007) whilst low body mass index is linked with increased all-cause mortality [20,21].

Therefore, despite the adoption of these prediction scores, the ‘heart team’ plays a central role in determining suitable candidates for TAVI or SAVR. Thus, with advancements in TAVI systems and technical knowledge, the focus of heart teams has shifted from discussing the technicalities of the procedure to assessing the patient and the important prognostic variables not represented by the EuroSCOREII and STS scoring systems to ensure the selection of patients most likely to benefit from TAVI. 

These results emphasise the importance of appropriate antithrombotic therapy following a TAVI procedure, given the often high-risk group of patients that undergo TAVI. Moreover, given the high rates of bleeding observed post-TAVI, these data highlight the clinical need for antithrombotic strategies that are tailored towards minimizing bleeding risk.

## 3. Mechanisms of Thrombosis in TAVI

The mechanisms underlying thrombosis associated with TAVI are likely multifactorial. In this regard, a number of contributing factors potentially explaining the thrombotic risk associated with TAVI have been proposed. These include: (1) flow disturbances associated with prosthetic valve placement, (2) the introduction of a prothrombotic metallic frame, and (3) a co-existent prothrombotic tendency in an older, co-morbid population [22,23,24]. Emerging evidence suggests that the haemodynamic disturbances created at sites of valve implantation play a leading role in thrombus formation [22,23,24,25,26]. Indeed, clinical data has demonstrated that the majority of thrombi forming around TAVIs occur on the aortic side of the implanted valve, between the leaflet and stent. This is significant since deployment of the stent and bioprostethic valve displaces the native valve, thus creating a so-called neosinus and smaller native sinus (Figure 1).

Recent data indicates that flow stagnation in the neosinus is likely to be an important haemodynamic factor associated with a heightened thrombotic risk [28,29,30,31,32,33,34]. These data, coupled with the clinical observation that prosthesis type and the larger transcatheter heart valves are also important risk factors for TAVI-associated thrombosis, suggest that patient-specific valve geometries at the site of implantation are likely a key driver of flow stagnation patterns and thus thrombotic risk [35]. It is well established that flow stagnation results in coagulation reactions that ultimately yield the generation of thrombin and a fibrin-rich thrombus [15,16,17].

The placement of the metallic frame creates a further nidus for thrombus formation. Although not formally tested, it is likely that placement of the metallic frame results in endothelial damage that in the context of local perturbation of endothelial matrix components and haemodynamic disturbances creates a prothrombotic milieu. Akin to coronary stents, prior to the endothelialisation of the metallic frame, the exposed artificial surface acts as a substrate for platelet adhesion and activation, as well as the initiation of the contact activation coagulation system. Furthermore, it has recently been established that haemodynamic forces play a central role in mediating platelet activation [36,37]. Together, these alterations are likely to confound any prothrombotic state. In this context, there is also a clinically relevant rate of pre- and new-onset AF in TAVI cohorts, which further exacerbates the risk of thrombosis and stroke, necessitating the use of anticoagulation, which increases the risk of bleeding complications [38,39]. Despite significant variance in incidence between registries, a recent meta-analysis revealed that the incidence of AF in the context of TAVI approximates 10% [40]. Importantly, new-onset AF was associated with an increase in 30-day all-cause mortality and a significant increase in one-year all-cause death. More recently, a meta-analysis of 65 studies incorporating 43,506 patients found the rate of new-onset AF was the most frequent arrhythmia post-TAVI [41]. Indeed, 2641 patients developed new-onset AF, with an increasing mean prevalence of 11%, 14%, 14% and 25% during hospital stay, 30-day, one-year and two-year follow-up periods, respectively. Significantly, new-onset AF following TAVI increased the risk of death (relative risk: 1.61, 95% CI: 1.35 to 1.98, I^2^ = 47%) and cerebrovascular events (1.79, 95% CI: 1.24 to 2.64, I^2^ = 0%) accordingly [41]. 

In addition to the emergence of AF and the local prothrombotic milieu that occurs in the setting of TAVI, periprocedural stroke can occur as a consequence of the dislodgment of calcified valvular components upon insertion of the prosthesis [42,43,44]. Embolic protection devices have been tested in association with TAVI. Although not analysed yet in large randomised trials, initial data suggest a reduction of strokes with the use of embolic protection devices [45,46]. 

## 4. Leaflet Thrombosis

Recent data has demonstrated that valve leaflet thrombosis is a common occurrence post-TAVI, with this complication being diagnosed in approximately 10–15% of patients by computed tomography angiography post-TAVI [47]. Interestingly, data from two registries from observational studies, SAVORY (Subclinical Aortic Valve Bioprosthesis Thrombosis Assessed with Four-Dimensional Computed Tomography) and RESOLVE (Assessment of Transcatheter and Surgical Aortic Bioprosthetic Valve Thrombosis and Its Treatment with Anticoagulation), suggest that leaflet thrombosis is more strongly prevalent among TAVI cohorts than SAVR cohorts [48], with the cumulative incidence of symptomatic leaflet thrombosis following TAVI ranging from 0.61% to 2.8% [49,50]. This supports the notion that, as discussed above, the TAVI procedure is associated with distinct haemodynamic changes that result in a prothrombotic phenotype. Whilst it is well accepted that subclinical leaflet thrombosis is a common occurrence, the clinical relevance of valve leaflet thrombosis remains a matter of much debate. Despite the detection of hypoattenuated leaflet thickening, a surrogate of valve leaflet thrombosis often being detected incidentally on computed tomography in patients without symptoms, the formation of valve leaflet thrombosis has been linked to two adverse sequelae, namely haemodynamic valve deterioration and an increased risk of stroke [51].

To date, the available evidence suggests that leaflet thrombosis can affect aortic valve pressure gradients; however, the magnitude of this effect ranges from clinically insignificant leaflet restriction to symptomatic valve deterioration [50]. In this regard, registry data from a cohort of 1521 TAVI patients with systematic clinical and echocardiography follow-up demonstrated that the rate of haemodynamic valve deterioration was 2.8% at one year post-TAVI [52]. Importantly, in all these cases, the underlying cause of valve deterioration was identified as valve thrombosis. Crucially, the rate of valve leaflet thrombosis appears to correlate with the antithrombotic regime such that patients managed with oral anticoagulation have a lower incidence of leaflet thrombosis. Moreover, it has consistently been shown that the treatment of patients with subclinical leaflet thrombosis with oral anticoagulation leads to a regression of valve thickening and is correlated with changes in transvalvular pressure gradients [53].

Interestingly, recent data suggest that the development of subclinical leaflet thrombosis following TAVI is not associated with increased mortality or stroke [54]. This observational study followed 754 patients (120 diagnosed cases of leaflet thrombosis), with no differences observed in all-cause mortality or the incidence of stroke/transient ischaemic attack (TIA) during a 406-day period (8-month Kaplan-Meier estimate for mortality: 86.6% vs. 85.4%, *p* = 0.912; for stroke- or transient ischaemic attack-free survival 98.5% vs. 96.8%, *p* = 0.331). In contrast, data from the MAUDE database highlights that leaflet thrombosis complicated by structural valve disease (SVD) (aortic stenosis or regurgitation) is associated with adverse outcomes including stroke, cardiogenic shock and death [55]. Indeed, of the 30 patients with leaflet thrombosis and SVD, stroke/TIA occurred in 10%, cardiogenic shock in 6.7% and death in 30%. These data emphasise that unlike subclinical leaflet thrombosis, clinically manifest leaflet thrombosis is associated with poor clinical outcomes.

## 5. Rationale for Optimised Therapy

With the inherent risk of stroke and leaflet thrombosis in the setting of TAVI, post-TAVI antithrombotic therapy is essential to minimise these risks. However, all currently clinically available antithrombotic therapy is invariably associated with a risk of bleeding [56]. In the context of TAVI, this is a particular issue since patients undergoing TAVI often have multiple co-morbidities, such as chronic kidney disease, that are well known to further increase bleeding risk [35]. Importantly, bleeding post-TAVI is associated with adverse clinical outcomes and increased mortality, thus diminishing the benefits of antithrombotic therapy [16]. Therefore, delineating the optimal antithrombotic therapy and duration of treatment remains an unmet clinical need. To date, the evidence base guiding antithrombotic therapy post-TAVI is weak, which consequently has yielded guideline recommendations that are not uniform across professional bodies. In addition to determining the optimal antithrombotic regime, identifying high-risk patients such as those with concurrent AF who stand to benefit from more intense antithrombotic regimens and the application of novel antithrombotics, which are not associated with untoward bleeding risk, remain two outstanding clinical issues in the context of TAVI.

## 6. Antithrombotic Therapy Guidelines Post-TAVI

The current guidelines regarding the use of antithrombotic therapy post-TAVI are summarised in Table 2. As previously discussed, these guidelines are predominantly based on small studies and consensus opinion. This has led to varying and largely empirical recommendations from different professional bodies, which is reflected in the heterogeneous antithrombotic regimes used in clinical practice. Whilst the development of antithrombotic therapies post-TAVI have no doubt been informed by the experience of SAVR with bioprostheses and PCI, given the very distinct nature of the TAVI procedure, metallic stent residual calcified valves and the effects of haemodynamics, it is obvious that more data is required to formulate adequate and uniform TAVI-specific guidelines.

## 7. The Role of Dual Antiplatelet Therapy

Current recommendations comprise administering dual antiplatelet therapy (DAPT), with clopidogrel used for the first 3–6 months and aspirin maintained indefinitely, given there is no indication for anticoagulation (Table 2). The therapeutic benefit of DAPT following TAVI for patients in sinus rhythm, however, remains contentious. Indeed, a pooled analysis investigating 672 participants comparing DAPT (aspirin and clopidogrel) vs. single antiplatelet therapy (SAPT) (aspirin) following TAVI demonstrated no significant discrepancy in the 30-day net adverse clinical and cerebral event (NACE) rate at one month [61]. The results indicated a NACE rate of 13% and 15% in the SAPT and DAPT groups respectively (odds ratio (OR): 0.83, 95% confidence interval (CI): 0.48 to 1.43, *p* = 0.50). A pattern, however, was noted in favour of fewer life-threatening and major bleeds in the ASA (aspirin) group compared with DAPT, questioning the value of clopidogrel in this context. Two recent meta-analyses conducted by Hu et al. and Ahmad et al. reached similar conclusions, suggesting that DAPT, when compared with SAPT following TAVI, displays non-superiority in reducing thrombotic events but involves added risk of major and life-threatening bleeds [62,63]. However, these findings are diluted somewhat by the fact that the analyses are based on small studies, many of which are non-randomised, with short follow-up periods. This is highlighted by a recent survey of 45 TAVI centres across Europe, Australasia and the USA, which revealed that 82.2% of the centres surveyed routinely use DAPT post-TAVI for a median of three months [63].

## 8. The Role of Anticoagulation Post-TAVI

The role of anticoagulation post-TAVI is vitally important but is still a highly contentious issue. This has come into focus with the observation that oral anticoagulation (OAC) may prevent subclinical leaflet thrombosis and valve deterioration [64]. Indeed, the 2017 American Heart Association guidelines recommend that anticoagulation with VKA to achieve an INR of 2.5 may be reasonable for at least three months following TAVI in patients at low risk of bleeding [58]. This recommendation is based on studies indicating the increased prevalence of subclinical valve thrombosis, as visualised by multidetector computerised tomographic scanning, in patients administered antiplatelet therapy alone versus anticoagulation with VKA [65]. However, as discussed above, the clinical relevance of subclinical leaflet thrombosis (SLT) remains to be established. More significant is the fact that AF is present in up to one-third of patients undergoing TAVI and, importantly, AF is associated with increased cardiovascular morbidity and mortality post-TAVI as a result of an elevated risk of stroke and bleeding related to OAC [38,39]. Indeed, data from the PARTNER trial emphasises this point, as patients with AF had a doubling of mortality at 12 months (26.2%) compared to those in sinus rhythm (12.9%) [66]. Strikingly, the mortality of patients with AF who experienced a major bleeding complication was 48.7%. These data are similar to previous reports in which patients with AF post-TAVI have been demonstrated to have a marked increase in mortality and significantly higher rates of bleeding [38,39]. 

Current guidelines recommend OAC (VKA or novel oral anticoagulant (NOAC)) in addition to aspirin for patients undergoing TAVI with concurrent AF. This practice has recently been supported by results from the FRANCE TAVI registry, where anticoagulation was correlated with decreased risk of bioprosthetic valve dysfunction after TAVI [67]. However, the benefit of combination therapy, as opposed to OAC alone, has been challenged recently. Indeed, single agent VKA appears as effective as VKA in combination with antiplatelet therapy, with the rates of stroke, adverse cardiovascular events and death similar between patients treated for a median of 13 months post-TAVI with VKA alone or VKA in combination with SAPT or DAPT [68,69,70]. Again, the use of antiplatelet agents in addition to VKA conferred a two-fold increase in the risk of major or life-threatening bleeding [68]. The notion that more intensive antithrombotic regimens may not translate to improved clinical outcomes has previously been highlighted by the WOEST study, where OAC plus clopidogrel in the setting of PCI was shown to be associated with significantly lower rates of bleeding and a reduction in all-cause mortality as compared with triple therapy (OAC in addition to aspirin and clopidogrel) [71].

Recently, the question as to whether NOACs may confer a lower bleeding risk compared to VKA in the setting of TAVI has begun to be addressed. A consistent finding from the large-scale trials evaluating NOACs for the prevention of arterial thromboembolism in AF and treatment of venous thromboembolism (VTE) has been the lower bleeding risk, especially intracranial bleeding, associated with NOACs in comparison to VKA [69,72]. Recently, a small study demonstrated that apixaban may allow anticoagulation post-TAVI, albeit with a lower bleeding risk compared to VKA [73]. However, recently, the much-anticipated GALILEO trial was stopped prematurely after interim analysis demonstrated a rivaroxaban-based antithrombotic regimen, compared to an antiplatelet regime, was associated with a higher incidence of thromboembolism, all-cause death and bleeding [74]. Whilst a full analysis of the data is awaited, it is important to note that the dose of rivaroxaban used in this trial was 10 mg, a dose which currently only has evidence for efficacy as a VTE thromboprophylaxis and for the secondary prevention of VTE. This raises the obvious question as to whether the dose of rivaroxaban was sufficient in the GALILEO trial. However, the issue of NOACs and their efficacy in the context of TAVI remains an area of active investigation with a number of ongoing trials (Table 3).

## 9. Ongoing Trials

The POPular-TAVI (antiPlatelet therapy for Patients undergoing Transcatheter Aortic Valve Implantation) trial is a multicentre open-label randomised trial currently investigating the hypothesis that monotherapy with aspirin or OAC is safer than dual therapy with clopidogrel for three months. Cohort A comprises patients on aspirin (up to 100 mg OD) versus aspirin plus clopidogrel (75 mg OD). Cohort B comprises patients on OAC (according to their indication) versus OAC plus clopidogrel. The primary outcome evaluated is freedom from non-procedure-related bleeding at one year and the trial will also assess for net clinical benefit [75].

The ATLANTIS (Anti-Thrombotic Strategy After Trans-Aortic Valve Implantation for Aortic Stenosis) trial is a Phase IIIb multicentre open-label randomised trial investigating the efficacy of apixaban following TAVI, compared with DAPT, for 12 months [76]. The experimental arm involves apixaban 5 mg twice daily, or 2.5 mg for patients over 80 years of age or body weight under 60 kg, irrespective of them having AF. This is compared with either VKA for patients with AF or standard of care antiplatelet therapy for sinus rhythm. 

The ENVISAGE-TAVI (Edoxaban Versus Standard of Care and Their Effects on Clinical Outcomes in Patients Having Undergone Transcatheter Aortic Valve Implantation (TAVI) in Atrial Fibrillation) trial is a multicentre open-label randomised trial comparing edoxaban to VKA-based therapy in patients with an indication for OAC after TAVI for up to 36 months [77]. The primary outcome evaluated is net adverse events and major bleeding. Patients are randomised to either 60 mg edoxaban OD or any VKA. Antiplatelet therapy may also be administered if indicated. 

The AUREA (Dual Antiplatelet Therapy Versus Oral Anticoagulation for a Short Time to Prevent Cerebral Embolism After TAVI) and AVATAR (Anticoagulation Alone Versus Anticoagulation and Aspirin Following Transcatheter Aortic Valve Interventions) trials are also smaller-scale multicentre randomised studies comparing VKA with DAPT and aspirin plus VKA therapy respectively for strokes and other complications [78,79].

## 10. Insights on Future Research

The ongoing trials discussed will provide important new insights regarding how NOACs may fit in the antithrombotic paradigm post-TAVI. However, a number of important challenges remain. The bleeding risk associated with current antithrombotic regimes remains a significant concern. In this regard, there have been a number of new antithrombotic drugs developed in the last decade that hold the promise of inhibiting thrombosis without significantly impeding haemostasis [56,81]. Foremost amongst these are inhibitors of protein disulfide isomerase (PDI), a novel PAR4 antagonist, and inhibitors of the intracellular platelet enzyme PI3K beta [82,83,84]. Both PDI inhibitors and PAR4 inhibitors are in phase 2/3 clinical trials and if proven safe and effective, could pave the way to a new range of antithrombotic drugs to be investigated for the prevention of TAVI-associated thrombosis. 

Another aspect of TAVI-associated thrombosis that is yet to be tested is whether mechanistically, the metallic frame associated with the valve causes significant contact factor activation. The RE-ALIGN trial, which investigated dabigatran for the prevention of thrombosis in patients with mechanical heart valves, had to be ended prematurely due to the elevated risk of bleeding and thrombosis associated with dabigatran [85]. Subsequent in vitro data suggest that in the context of mechanical heart valves, contact factor activation plays a predominant role instigating thrombin generation, which may explain why warfarin is superior to a direct thrombin inhibitor in this context [86]. Therefore, whether similar effects in the haemostatic system occur in the context of TAVI requires investigation. Significantly, a novel anti-XIIa antibody has recently been developed that exhibits marked potency for inhibiting contact factor activation without any effect on haemostasis and could thus provide scope for another novel, potentially safer antithrombotic approach post-TAVI [87].

As discussed above, a major outstanding issue is identifying the patients most likely to benefit from more intense antithrombotic strategies post-TAVI. Our group has developed novel therapeutic and imaging agents based on a unique single-chain antibody that specifically binds to activated platelets [88]. Thus, this technology could be utilised to image patients for the specific and sensitive detection of valve thrombosis using ultrasound, CT or MRI modalities [88,89]. Intensified antithrombotic therapy could then only be used in patients where thrombus is detected. Alternatively, this approach would provide the potential to administer platelet-targeted anti-thrombotic therapy, which we have previously demonstrated to spare haemostasis [90].

In addition to novel therapeutics, there remains much interest within the bioengineering field to design and develop new valves that may produce fewer haemodynamic disturbances and expose less thrombogenic material and as such, may result in a lower thrombosis risk [91]. 

## 11. Conclusions

The introduction of TAVI represents one of the most successful innovations in cardiovascular medicine in the last two decades. This has afforded patients with severe aortic stenosis, especially the elderly and those with significant co-morbidities, access to treatment where surgery is often contraindicated. With significant improvements in mechanical durability and implantation technologies, TAVI-related thromboses and bleeding associated with antithrombotic therapy, are becoming the major determinants of the long-term outcome of TAVI. A growing recognition of the shortcomings of the current antithrombotic regimes used to prevent TAVI-associated thrombosis has led to several ongoing clinical trials, which will no doubt inform the development of new practice guidelines. Simultaneously, with a deepening understanding of the mechanisms causing the development of post-TAVI thrombosis and the recent advent of a range of novel antithrombotics that do not inhibit haemostasis, the stage is set for the development of safer and more effective TAVI-specific antithrombotic therapy.

## Figures and Tables

**Figure 1 jcm-08-00280-f001:**
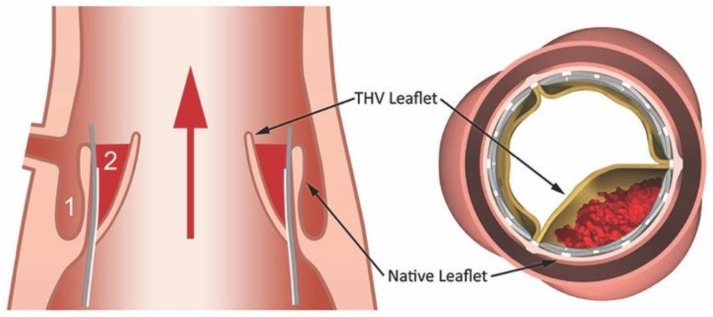
The native sinus and neosinus. Deployment of the transcatheter heart valve (THV) results in the displacement of the native aortic valve leaflets. The displaced native valve leaflets divide the aortic sinus into two parts—a neosinus (1) and native sinus (2). Flow stagnation in the neosinus appears to be a key driver of thrombus formation in the context of TAVI. Reproduced with permission from Yoganathan A, The Fluid Mechanics of Transcatheter Heart Valve Leaflet Thrombosis in the Neosinus; published by Lippincott Williams & Wilkins, 2017 [27].

**Table 1 jcm-08-00280-t001:** Ischaemic and bleeding complications in post-transcatheter aortic valve implantation (TAVI) patients [3,11,12,13,14].

	<30 Days	1 Year	2 Years
Life-threatening/major bleeding complications (%)	10.2 ± 3.5	15.95 ± 0.9	17.6 ± 0.7
Stroke (%)	4.1 ± 0.7	7 ± 1.7	8.5 ± 2.3
Disabling stroke (%)	2.4 ± 1.3	4.1 ± 1.8	4.9 ± 2.1
New-onset atrial fibrillation (AF) (%)	11.2 ± 1.9	13 ± 4.1	15.4 ± 5.7
Myocardial infarction (%)	0.9 ± 0.1	2.1 ± 0.3	2.7 ± 0.8
All-cause mortality (%)	2.8 ± 0.6	10.3 ± 3.7	15.9 ± 5.6
Cardiovascular mortality (%)	2.6 ± 0.6	7.1 ± 2.9	10.7 ± 4.1

Data presented as the number of events divided by the number of treated patients, using available data from the Surgical Replacement and Transcatheter Aortic Valve Implantation (SURTAVI) trial, the Placement of Aortic Transcatheter Valves (PARTNER) 2 cohort and the US CoreValve High Risk study. Results are presented as weighted mean ± 1 standard deviation. Bleeding complications: major or life-threatening bleeding complications; new-onset AF: new-onset atrial fibrillation.

**Table 2 jcm-08-00280-t002:** Current antithrombotic guideline recommendations for patients undergoing TAVI.

	American College of Cardiology/American Heart Association (AHA)/Society of Thoracic Surgeons [57,58]	European Society of Cardiology (ESC) [59]	American College of Chest Physicians [60]
TAVI Post-Procedural	75–100 mg aspirin OD indefinitely	Aspirin or clopidogrel indefinitely	50–100 mg aspirin OD indefinitely (Grade 2C)
	75 mg clopidogrel OD for 6 months	Aspirin and clopidogrel early post-TAVI	75 mg clopidogrel OD for 3 months (Grade 2C)
	If VKA indicated, no clopidogrel	If VKA indicated, no antiplatelet therapy	
Bioprosthetic valves			
Low risk	75–100 mg aspirin OD (Class IIaB ^a^)	Low-dose aspirin (Class IIaC ^b^)	50–100 mg aspirin OD indefinitely (Grade 2C)
	VKA (target INR 2.5) for at least 3 months (Class IIbB ^b^)	VKA (target INR 2.0–3.0) (Class IIbC ^c^)	
High risk	75–100 mg aspirin OD (Class IIaB ^a^)	VKA (target INR 2.5) (Class IC ^a^)	
	VKA (target INR 2.0–3.0) (Class I ^a^)		

AHA risk factors: new-onset atrial fibrillation (AF), left ventricular dysfunction, previous thrombo-embolism, and hypercoagulable condition; ESC risk factors: AF, venous thrombo-embolism, hypercoagulable state, or with a lesser degree of evidence, severely impaired left ventricular dysfunction (ejection fraction ≤35%). OD: once daily; AF: atrial fibrillation; INR: international normalised ratio; TAVI: transcatheter aortic valve implantation; VKA: vitamin K antagonist. [14]. ^a^ Class I: conditions for which there is evidence for and/or general agreement that the procedure or treatment is beneficial, useful and effective. ^b^ Class IIa: weight of evidence/opinion is in favour of usefulness/efficacy. ^c^ Class IIb: usefulness/efficacy is less well established by evidence/opinion.

**Table 3 jcm-08-00280-t003:** Ongoing trials with novel oral anticoagulants in patients post-TAVI.

	POPular-TAVI	ATLANTIS	ENVISAGE-TAVI AF	AUREA	AVATAR
(ClinicalTrials. gov) Identification	NCT02247128	NCT02664649	NCT02943785	NCT01642134	NCT02735902
Study Design	Multicentre, open-label, randomised	Multicentre, open-label, randomised, phase IIIb	Multicentre, open-label, randomised, phase IIIb	Multicentre, randomised, phase IV	Multicentre, open-label, randomised
Patient Cohort	No need for long-term OAC	Successful TAVI without consideration of previous antithrombotic treatment	Successful TAVI. Patients have AF and an ongoing indication for OAC.	Patients with successful TAVI procedure not under OAC treatment.	Successful TAVI procedure and patient receiving VKA prior to procedure
Experimental Drug	Cohort A: 75 mg clopidogrel OD and <100 mg aspirin OD. Cohort B: 75 mg clopidogrel and OAC.	5 mg apixaban. 2.5 mg apixaban, if the patient has 2 or more factors ^a^	Edoxaban-based regimen 60 mg and 30 mg tablet OD and 15 mg tablet for transitioning at end of treatment. Antiplatelet therapy (if prescribed): aspirin 75–100 mg OD or clopidogrel 75 mg OD	VKA (acenocumarol)	VKA (target INR 2–3)
Comparator	Cohort A: <100 mg aspirin. Cohort B: OAC.	VKA if AF or antiplatelet therapy if sinus rhythm	VKA-based regimen oral VKA tablets as selected and provided by the site. (Target INR 2–3)	DAPT with 100 mg aspirin and 75 mg clopidogrel	75–100 mg aspirin and VKA (VKA corresponding to anticoagulation the patient was receiving prior to TAVI, monitored and adapted to current recommendations)
Primary Outcome	Lack of bleeding complications as per BARC definition 1 year post-TAVI. Co-primary outcome defined as freedom of non-procedure-related bleeding complications at 1 year post-TAVI.	Composite of death, MI, stroke, peripheral embolism, intracardiac or bioprosthesis thrombus, any episode of DVT or PE, major bleeding (up to 13 months)	Number of participants experiencing any of these factors ^b^ (within 36 months). Number of participants experiencing major bleeding (within 36 months).	Cerebral thromboembolism by the detection of cerebral infarction by MRI within the first 3 months post-TAVI (within 3 months).	Composite of all-cause death, MI, stroke, valve thrombosis and haemorrhage >2 as defined by the VARC 2. (within 12 months).
Duration	Aspirin for at least 12 months, with a lifelong recommendation. Clopidogrel discontinued after 90 days in both cohorts	12 months	VKA continued until efficacy endpoints are reached or up to 36 months post-randomisation. Aspirin or clopidogrel discontinued after 90 days. Patients with stenting post-TAVI continue aspirin or clopidogrel up to 12 months, DAPT allowed for 1 month.	3 months	12 months
Estimated Enrolment	1000	1509	1400	124	170

OAC: oral anticoagulation; DAPT: dual antiplatelet therapy; VARC: valve academic research consortium; BARC: bleeding academic research consortium; INR: international normalised ratio; MI: Myocardial infarction; PE: pulmonary embolism; DVT: deep vein thrombosis; MRI: Magnetic resonance imaging [75,76,77,78,79,80]. ^a^ age >80years, body weight <60 kg, serum creatinine >1.5 mg/dL/; ^b^ all-cause death, MI, ischaemic stroke, valve thrombosis, and major bleeding.

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
