# Peer review of "Thromboembolic and Bleeding Complications in Transcatheter Aortic Valve Implantation: Insights on Mechanisms, Prophylaxis and Therapy"

_jcm, 2019, doi:10.3390/jcm8020280_

Reviewer 1 Report

Ranasinghe and colleagues have provided an excellent summary of contemporary issues surrounding the use of anti-platelets and anti-thrombotics in patients undergoing TAVI.

The review is detailed, and covers all relevant current evidence.

Overall, the review is very well written. They have covered all major current clinical issues and given a good summary of the ongoing studies.

I do not have any major concerns with regards to the review.

Author Response

Dear Reviewer,

Thank you kindly for your feedback and taking the time to review our work.

We thank you for your positive and highly constructive comments.

Kind regards,

Karlheinz Peter

Reviewer 2 Report

Thank you for submitting this excellent review to the Journal of Clinical Medicine. I was pleased to receive it as a reviewer. This review highlights the major challenge of post-TAVI thrombosis and bleeding, and the significant issues surrounding  current antithrombotic approaches. I don’t have questions for you to be addressed due to the interesting topic. Good luck with your paper, and thanks again for submitting it.

Author Response

(The authors gave the same response as above.)

Reviewer 3 Report

The authors present in this excellent overview current state of art of anticoagulation management after TAVR procedures. All new relevant data are included, as well as the ongoing prospective trials on this topic. I have no major comments, the authors have to be congratulated for their outstanding contribution  

Author Response

(The authors gave the same response as above.)
